# NESTOR: A NESTED MOE-BASED NEURAL OPERATOR FOR LARGE-SCALE PDE PRE-TRAINING

## ABSTRACT

Neural operators have emerged as an efficient paradigm for solving PDEs, overcoming the limitations of traditional numerical methods and significantly improving computational efficiency. However, due to the diversity and complexity of PDE systems, existing neural operators typically rely on a single network architecture, which limits their capacity to fully capture heterogeneous features and complex system dependencies. This constraint poses a bottleneck for large-scale PDE pre-training based on neural operators. To address these challenges, we propose a large-scale PDE pre-trained neural operator based on a nested Mixture-of-Experts (MoE) framework. In particular, the image-level MoE is designed to capture global dependencies, while the token-level Sub-MoE focuses on local dependencies. Our model can selectively activate the most suitable expert networks for a given input, thereby enhancing generalization and transferability. We conduct large-scale pre-training on twelve PDE datasets from diverse sources and successfully transfer the model to downstream tasks. Extensive experiments demonstrate the effectiveness of our approach.

## 1 INTRODUCTION

Partial differential equations (PDEs) have broad applications in science and engineering, including physics and fluid mechanics Karniadakis et al. (2021) Debnath (2005). Existing studies can be roughly divided into two categories: traditional numerical methods and data-driven methods. Traditional methods, such as FEM Norrie & De Vries (2014) and FDM LeVeque (2007), approximate PDE solutions by discretizing the spatial domain, resulting in complex procedures and high computational costs. Neural operators aim to learn infinite-dimensional mappings between function spaces Li (2021), enabling fast inference while maintaining reasonable accuracy, significantly reducing computational costs, and overcoming the limitations of traditional methods. However, neural operators typically rely on large amounts of training data, which are often obtained through costly experiments and numerical simulations, severely limiting their application in wider scenarios.

Recently, large-scale pre-training Bengio (2012) offers a new research paradigm to address this problem. Unlike traditional methods, it involves initially training models on large-scale datasets, enabling them to acquire generalizable knowledge across different PDEs and tasks, thereby establishing a unified modeling framework. For specific downstream tasks, only a small amount of data is required for fine-tuning to obtain highly accurate solutions. This paradigm not only enhances model generalization and effectively mitigates overfitting but also significantly reduces the training cost and time for downstream tasks. Large-scale pre-training has been widely applied in fields such as computer vision and natural language processing Dosovitskiy et al. (2020) Devlin et al. (2019), where its superior performance has been well validated in practice.

In the field of neural operators Lu et al. (2019) Li et al. (2020), research on large-scale pre-training for PDEs has begun to take shape Hao et al. (2024). However, PDE systems are highly complex, not only involving multiple types of equations but also containing physical fields with intricate spatio-temporal dependencies and regional similarities, resulting in complex data distributions and highly diverse tasks. Existing approaches typically use a single network architecture. Although such models can capture general knowledge of equations, they are limited in representing the specific characteristics of different types of PDE and the regional correlations within the physical fields of each equation, as shown in Fig. 1. If a model can finely learn the unique properties of a particular

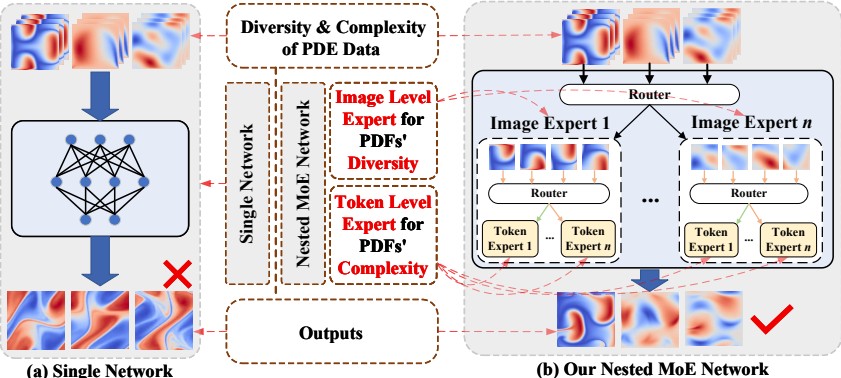

Figure 1: Comparison of two different network architectures. (a) Traditional single-network architecture; (b) our proposed nested MoE architecture, where image-level MoE experts learn global diversity across different PDE types, while token-level Sub-MoE experts capture complex local features within equations.

class of equations and effectively identify both local and global correlations in physical fields, its generalization and cross-task transfer performance can be significantly enhanced. In recent years, the Mixture-of-Experts (MoE) framework Jacobs et al. (1991) has attracted significant attention due to its advantages in increasing model capacity while maintaining computational efficiency. Through a routing mechanism Jacobs et al. (1991), the MoE selectively activates certain expert networks, choosing the most suitable experts for each input, providing a new research idea for large-scale pre-training of PDE neural operators. However, although single-layer MoE models can capture feature differences between equation types, they still face limitations in modeling diversity and complexity within physical fields of the same type of equations.

To address these challenges, we innovatively incorporate the MoE architecture into our model design, constructing a **NEST**ed MoE-based neural **O**perato**R** for large-scale PDE pre-training (**NESTOR**). Specifically, we first design a series of image-level MoE experts to learn the global diversity of a class of PDEs and adaptively activate the most suitable expert through image-level routing to process inputs of similar PDE types. Within each image-level expert, multiple token-level sub-MoE experts are set up to further capture the complex local dependencies of the physical field in the equation, and selectively activate the most suitable experts through token-level routing for processing. This nested MoE architecture solves the problems of PDE diversity and complexity from two levels. Through pre-training on large-scale PDE datasets, this architecture is successfully transferred to downstream tasks, providing an efficient solution for complex PDE problems. The main contributions of this work can be summarized as follows:

- Proposed a nested MoE framework. We design a novel nested MoE architecture that integrates image-level MoE and token-level MoE within a unified framework, enabling cross-level expert collaboration.
- Designed an image-level routing mechanism. We develop an image-level routing mechanism that selects appropriate expert networks based on the global characteristics of the data, providing a holistic perspective.
- Comprehensive validation on large-scale PDE datasets. We apply the proposed framework to large-scale pre-training and downstream tasks across multiple PDE datasets, demonstrating significant advantages in cross-task generalization and transferability.

## 2 RELATED WORKS

### 2.1 NEURAL OPERATORS

Neural operators are designed to learn mesh-free, function-space-to-function-space infinite-dimensional mappings from inputs to solution functions Lu et al. (2019). They effectively overcome the dependence of traditional numerical solvers on mesh discretization, improving computational

speed and reducing costs. Moreover, for repeated problems, a neural operator only needs to be trained once, without retraining for each new PDE instance, making it an efficient paradigm for PDE solving. To successfully apply neural operators to PDE problems, researchers have proposed several effective model architectures. For example, DeepONet Lu et al. (2019) adopts a branch–trunk architecture to realize operator learning. The Fourier Neural Operator (FNO) Li et al. (2020) leverages Fourier transforms to capture non-local dependencies, thus enabling efficient PDE solutions. The Galerkin Transformer Cao (2021) integrates self-attention mechanisms with Galerkin projection for operator learning. GNOT Hao et al. (2023) combines graph neural operators with Transformers, achieving efficient modeling on irregular meshes. MPP McCabe et al. (2023) is a Transformer-based autoregressive pre-training architecture. DPOT Hao et al. (2024) employs autoregressive denoising pre-training combined with Fourier attention to predict a wide range of PDE problems. Despite the significant progress made by neural operators, their performance still has room for improvement due to the limitations imposed by the diversity of data and tasks.

## 2.2 MIXTURE OF EXPERTS

The Mixture of Experts (MoE) framework is a method that expands model capacity while avoiding a significant increase in computational cost. Its core idea is to select a subset of experts among multiple expert networks through a gating mechanism Jacobs et al. (1991). With the development of MoE, it has been widely applied in natural language processing, computer vision, and other domains. GShard Lepikhin et al. (2020) was the first to introduce the MoE structure into Transformer models, enabling efficient large-scale distributed training. Switch Transformer Fedus et al. (2022) scaled large language model parameters to the trillion level, significantly improving both model capacity and efficiency. V-MoE Riquelme et al. (2021) applied MoE to vision Transformers and demonstrated its potential for enhancing efficiency and performance in tasks such as image recognition. Existing work primarily focuses on homogeneous experts, while research on heterogeneous Wang et al. (2024) experts is relatively limited. Homogeneous experts refer to all experts using the same network architecture, which offers simplicity in implementation, stable convergence, and ease of load balancing. However, having identical architectures limits expert diversity and, to some extent, constrains the performance of MoE. Heterogeneous expert MoE allows different experts to adopt different network architectures, avoiding redundancy in the features learned by the experts and significantly enhancing the model's expressive power and efficiency.

## 2.3 PRE-TRAINING

Pre-training Bengio (2012) refers to the process of training a model on large-scale datasets to learn general knowledge that can be transferred to a variety of downstream tasks. It can significantly reduce the training cost of downstream tasks while improving generalizability. The pre-training paradigm has achieved outstanding success in natural language processing, demonstrating strong cross-task transferability, as exemplified by models such as BERT Devlin et al. (2019) and GPT Radford et al. (2018). In computer vision, pre-training has also been widely adopted, with notable examples including the Vision Transformer (ViT) Dosovitskiy et al. (2020) and CLIP Radford et al. (2021). With the development of large-scale pre-training models, this approach has gradually been introduced into the field of PDE neural operators. Existing explorations include MPP McCabe et al. (2023), which proposes a Transformer-based autoregressive pre-training framework capable of learning unified serialized representations across various PDE datasets and allowing cross-task modeling through transfer. DPOT Hao et al. (2024) employs an autoregressive denoising strategy combined with Fourier attention to achieve efficient pre-training across multiple types of PDE problems, demonstrating cross-equation generalization at the operator level. Although these studies have successfully applied pre-training techniques to PDE neural operators, they still exhibit notable limitations in comprehensively capturing PDE systems. Therefore, there remains substantial room for further exploration of large-scale pre-training in the PDE neural operator domain.

## 3 PROPOSED METHOD

Our NESTOR model aims to address the PDEs' diversity and complexity from image and token levels. This section starts with an overview of the proposed NESTOR model. Then we provide a detailed description of each part of the model. Finally, the loss function is presented.

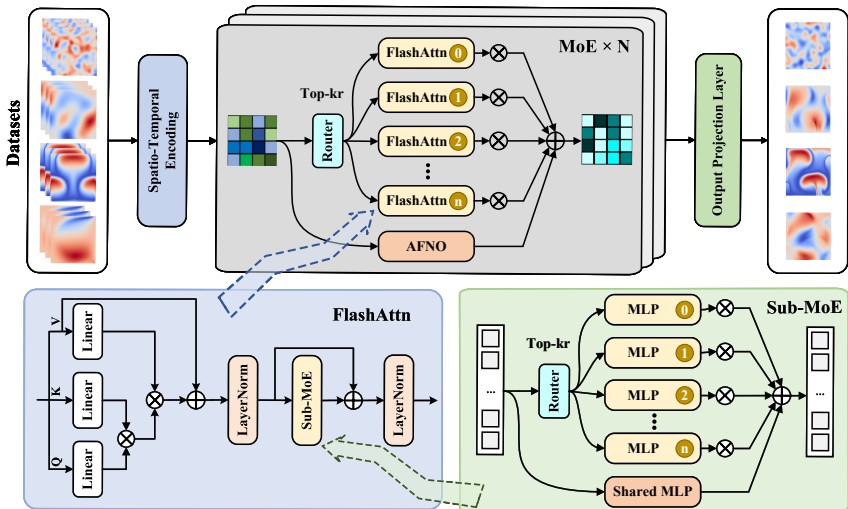

Figure 2: Overview architecture. We trained on twelve mixed PDE datasets, predicting the next frame based on the preceding frames. We designed a nested MoE architecture: (1) the top shows the overall model architecture; (2) the bottom right illustrates the nested Sub-MoE architecture; and (3) the bottom left depicts the improved FlashAttention architecture.

## 3.1 OVERVIEW

In this paper, we consider the general form of a parameterized partial differential equation defined on the spatial region $\Omega \subset \mathbb{R}^n$ and the time interval $[0, T]$,

$$\frac{\partial u}{\partial t} - \mathcal{F}\big(u, \nabla u, \nabla^2 u, \ldots; \theta\big) = 0, \tag{1}$$

$$\begin{cases} u(x, 0) = u_0(x), & x \in \Omega, \\ \mathcal{B}[u](x, t) = g(x, t), & (x, t) \in \partial\Omega \times (0, T], \end{cases}$$

where $u$ is the unknown solution function, representing the state of the system; $\mathcal{F}$ is the PDE spatial derivative operator, which describes the dynamics or evolution law of the system and depends on the current solution u, its spatial derivative, and parameter $\theta$; $\theta$ is the external condition or physical parameter that controls the properties of the equation; u(x,0) is the initial condition; $\mathcal{B}[u](x, t)$ is the boundary condition.

On this basis, we define a solution operator $\mathcal{F}$ and construct the following mapping

$$\mathcal{F}: \quad u_{t+1} = \mathcal{F}_T(u_{t-T+1:t}; \theta), \tag{2}$$

where $\theta$ represents the system parameters. Based on given conditions and parameters, the operator $\mathcal{F}$ can take the most recent T frames as input and predict the next frame from the previous T frames, thereby predicting the evolution of different system states.

When dealing with complex, high-dimensional continuous partial differential equations, Transformers struggle to effectively represent kernel integral operators Guibas et al. (2021). Meanwhile, traditional neural operators have difficulty fully capturing diverse data features and complex system dependencies. To address these challenges, we propose a nested MoE framework, as illustrated in Fig. 2. The model first maps the PDE inputs into a series of latent representations. These latent representations are then processed by the MoE module, where a learned gating mechanism assigns them to different experts, enabling each expert to learn distinct features of the inputs. The proposed network architecture adaptively captures multi-scale features of multiphysics fields, demonstrating strong transferability and cross-task generalization capability.

## 3.2 SPATIO-TEMPORAL ENCODING

First, the input $x \in \mathbb{R}^{B \times C \times H \times W}$ is divided into a set of non-overlapping patches $X_p \in \mathbb{R}^{B \times N \times C \times P_H \times P_W}$, where $B$ is the batch size, $N$ is the number of patches, and $(P_H \times P_W)$ is

the patch size. Each patch is then projected into a $D$-dimensional space, followed by the addition of positional encoding $E_{\text{pos}}$:

$$X = \text{Embedding}(X_p) + E_{\text{pos}} \in \mathbb{R}^{B \times N \times D}, \tag{3}$$

Subsequently, the obtained representation is rearranged as $X \in \mathbb{R}^{B \times X \times Y \times T \times C}$, and mapping time series to a fixed dimension to compress information in the time dimension:

$$Y = \sum_{t=1}^{T} W_t X_t, \quad Y \in \mathbb{R}^{B \times X \times Y \times C_{\text{out}}}, \tag{4}$$

where $W \in \mathbb{R}^{T \times C_{\text{out}} \times C_{\text{out}}}$ is a learnable weight matrix.

### 3.3 NESTED MIXTURE-OF-EXPERTS (NESTOR) ARCHITECTURE

A single type of network architecture is insufficient to fully capture the diverse characteristics of data. To address this, we introduce a nested MoE architecture at the operator level to enable multi-scale interactions within the PDE system. This module dynamically allocates the most appropriate expert network through a routing mechanism, allowing it to simultaneously characterize both local and global dependencies and effectively capture features in both the time and frequency domains.

#### 3.3.1 ROUTING STRATEGY

In the main MoE module, we adopt an image-level gating mechanism combined with a top-$k$ Shazeer et al. (2017) routing strategy for expert selection. The detailed process is as follows.

First, given the input feature $x \in \mathbb{R}^{B \times C \times H \times W}$, we apply global average pooling to obtain the image-level representation $\bar{x}_b \in \mathbb{R}^C$, where $b = 1, \ldots, B$. Next, the image-level representation is fed into a learnable linear layer to produce the raw expert scores:

$$s_b = \bar{x}_b W^\top + b \in \mathbb{R}^N, \tag{5}$$

where $W \in \mathbb{R}^{N \times C}$ is the expert weight matrix, $b \in \mathbb{R}^N$ is the bias term, and $N$ denotes the number of experts. The raw scores are then normalized using the softmax to obtain the routing probabilities

$$p_b = \text{softmax}(s_b), \quad \sum_{i=1}^{N} p_{b,i} = 1. \tag{6}$$

Finally, according to the top-$k$ routing strategy, the $k$ experts with the highest probabilities are selected. Let $\mathcal{I}_b$ denote the index set of the selected experts. For each selected expert $i \in \mathcal{I}_b$, the final routing weight is defined as:

$$w_{b,i} = \frac{p_{b,i}}{\sum_{j \in \mathcal{I}b} pb,j}, \quad i \in \mathcal{I}_b. \tag{7}$$

#### 3.3.2 EXPERT DESIGN

**1) Shared Expert.** In the main MoE module, we select AFNO Guibas et al. (2021) as the shared expert, which is primarily responsible for capturing cross-task global spatial low-frequency features. First, the input feature $x \in \mathbb{R}^{B \times C \times H \times W}$ is Fourier transformed: $\hat{x} = \mathcal{F}(x)$, $\hat{x} \in \mathbb{C}^{B \times H \times W \times C}$, where $\mathcal{F}(\cdot)$ represents the FFT operation. Next, a complex convolution operation is performed in the frequency domain

$$\hat{y}_{\text{real}} = \sigma\left( \hat{x}_{\text{real}} W_1^{(r)} - \hat{x}_{\text{imag}} W_1^{(i)} + b_1^{(r)} \right), \tag{8}$$

$$\hat{y}_{\text{imag}} = \sigma\left( \hat{x}_{\text{imag}} W_1^{(r)} + \hat{x}_{\text{real}} W_1^{(i)} + b_1^{(i)} \right), \tag{9}$$

where $\sigma$ is the activation function, $W_1^{(r)}, W_1^{(i)}$ are the learnable matrices for the real and imaginary parts, respectively, and $b_1^{(r)}, b_1^{(i)}$ are bias terms. Then, an inverse Fourier transform is performed to return to the spatiotemporal feature representation

$$y = \mathcal{F}^{-1}(\hat{y}), \tag{10}$$

where $\mathcal{F}^{-1}(\cdot)$ represents the IFFT operation. Finally, a normalization layer, MLP, and residual connections are combined to obtain the output of the shared expert.

**2) Non-shared Expert.** We design FlashAttention Dao et al. (2022) as a non-shared expert, applied to the image-level features after routing. Here, the standard FFN layer within FlashAttention is replaced by a Sub-MoE, which is primarily responsible for capturing dependencies among tokens within an image. First, the input feature $x \in \mathbb{R}^{B \times C \times H \times W}$ is reshaped into a sequence form $x' \in \mathbb{R}^{B \times C \times N}$, where $N = H \times W$. Next, $x'$ is normalized and linearly transformed to obtain the query ($Q$), key ($K$), and value ($V$) representations. The attention-weighted result is then computed as $Z = \text{softmax}\left(\frac{QK^{\top}}{\sqrt{d_k}}\right)V$, which is added to the input residual and further normalized to obtain $\tilde{Z}$. Subsequently, $\tilde{Z}$ is passed through a Sub-MoE module for linear transformation:

$$Y = \text{Sub-MoE}(\tilde{Z}). \tag{11}$$

Finally, by combining residual connections and normalization layers, we obtain the output of the non-shared expert.

### 3.3.3 SUB-MOE

**1) Routing Strategy.** In Sub-MoE, we adopt a token-level gating mechanism Fedus et al. (2022) combined with a top-k routing strategy for expert selection. Unlike the image-level gating in the main MoE module, the token-level gating computes expert scores for each token individually, enabling a finer-grained expert selection: $s_{b,n} = x_{b,n}W^{\top} + b$, $s_{b,n} \in \mathbb{R}^M$, where $x_{b,n} \in \mathbb{R}^C$ represents the $n$th token feature of the $b$th sample, $W \in \mathbb{R}^{M \times C}$ is the expert weight matrix, $b \in \mathbb{R}^M$ is the bias term, and $M$ is the number of experts. Afterward, the raw scores are normalized, and the $k$ expert index set $\mathcal{I}b, n$ with the highest scores is selected. Each token is then dynamically assigned to the selected expert for processing:

$$wb, n, i = \frac{p_{b,n,i}}{\sum_{j \in \mathcal{I}b,n} pb, n, j}, \quad i \in \mathcal{I}_{b,n}. \tag{12}$$

where $pb, n, i$ is the normalized score of the $n$th token in the $b$th sample for the $i$th expert, and $w_{b,n,i}$ is the final weight assigned.

**2) Sub-Expert Design.** Sub-MoE implements the functionality of the FFN layer in FlashAttention and is a homogeneous MoE. This means that both shared and unshared experts use the same network structure, designed as an MLP. Normalized features are fed into the Sub-MoE, where token-level routing assigns them to the most appropriate expert for processing, extracting fine-grained feature representations. The computational process is as follows.

$$\text{ExpertMLP}(x) = W_2 \, \sigma(W_1 x + b_1) + b_2, \tag{13}$$

where $W_1 \in \mathbb{R}^{C \times (rC)}$, $W_2 \in \mathbb{R}^{(rC) \times C}$, $r$ is mlp_ratio, $\sigma(\cdot)$ denotes the activation function of GELU. Specifically, we first perform the first-layer linear transformation on the input feature $h = xW_1 + b_1$. Next, perform a nonlinear activation on $h$: $a = \text{GELU}(h)$. Finally, a second linear transformation is performed to obtain the final feature representation: $y = aW_2 + b_2$.

### 3.4 HEAD AND LOSS FUNCTION

#### 3.4.1 LOAD BALANCING LOSS

In our nested MoE model, the routing mechanism assigns tokens to the most suitable experts. A balanced distribution of tokens among experts is crucial for MoE performance. When the allocation is imbalanced, some experts remain idle and fail to learn diverse features, while a few experts become overloaded, potentially causing memory bottlenecks. This can lead the model to degenerate to using only a subset of experts, failing to fully leverage the advantages of MoE. To address this issue, we introduce a load-balancing loss Shazeer et al. (2017) to encourage a more uniform distribution of tokens across experts. Here, the two load balancing losses are defined following the same pattern:

$$\mathcal{L}_{\text{aux}} = E \sum_{i=1}^{E} p_i \cdot f_i, \qquad p_i = \frac{1}{N} \sum_{j=1}^{N} P_{ij}, \qquad f_i = \frac{n_i}{\sum_{k=1}^{E} n_k}, \tag{14}$$

where $p_i$ is the routing probability of expert $i$, $f_i$ is the actual token assignment ratio of expert $i$, $E$ is the total number of experts, $N$ is the total number of tokens, $P_{ij}$ is the probability of token $j$ being assigned to expert $i$, and $n_i$ denotes the number of tokens assigned to expert $i$.

### 3.4.2 MAIN TASK LOSS

For our regression task, we choose $\mathcal{L}_2$ loss Li et al. (2020) as the main task loss function:

$$\mathcal{L}_2 = \frac{\left\| \hat{y}_i^{(c)} - y_i^{(c)} \right\|_2}{\left\| y_i^{(c)} \right\|_2}, \tag{15}$$

where $y_i^{(c)}$ is the ground-truth of $i$-th sample at channel $c$, and $\hat{y}_i^{(c)}$ is the corresponding prediction.

### 3.4.3 TOTAL LOSS

Ultimately, our loss function consists of the main task loss and two load-balancing losses:

$$\mathcal{L} = \mathcal{L}_2 + \alpha\mathcal{L}_{\text{aux1}} + \beta\mathcal{L}_{\text{aux2}}, \tag{16}$$

where $\mathcal{L}_2$ denotes the main task's $\mathcal{L}_2$ loss; $\mathcal{L}_{\text{aux1}}$ is the load balancing loss of the MoE module (image-level routing); $\mathcal{L}_{\text{aux2}}$ is the load balancing loss of the Sub-MoE module (token-level routing); and $\alpha$ and $\beta$ are hyperparameters that control the contribution of the load balancing losses.

## 4 EXPERIMENTS

### 4.1 DATASETS AND EVALUATION METRIC

**Datasets.** We conducted experiments on a mixed dataset consisting of twelve different data sources and different parameters from FNO Li et al. (2020), PDEBench Takamoto et al. (2022), PDEArena Gupta & Brandstetter (2022), and CFDBench Luo et al. (2023). (1) FNO: A dataset containing three different parameters for the same type of equation. (2) PDEBench: A dataset containing four different parameters for the same type of equation. (3) PDEArena: A dataset containing the same equation with and without initial conditions. It is worth noting that, due to certain reasons, the NS and NS-cond datasets are missing 1,300 and 604 samples, respectively. Our model is trained under this reduced-data setting, while the baseline models are trained on the complete datasets. (4) CFDBench: A multi-task PDE dataset obtained by processing the four subtasks uniformly.

**Evaluation Metrics.** We choose the relative error $\mathcal{L}_2$ as the evaluation metric, where lower relative error values $\mathcal{L}_2$ indicate better performance.

### 4.2 MAIN RESULTS

Table 1 presents the experimental results of our method compared with other models in the pre-training datasets. The first row of the table specifies the types of PDE datasets and parameter settings, while the first column lists the baseline models for comparison. The experiments are divided into two parts: the first is pre-training, where all models are trained from scratch on the datasets; the second is fine-tuning, where models are further trained based on the pre-trained weights.

In the pre-training stage, our method demonstrates strong performance across 12 PDE datasets, achieving state-of-the-art results on 6 of them. Notably, our model ranks first on 5 out of 6 PDEBench datasets, and achieves significantly lower errors than mainstream models on multiple benchmarks. These results clearly validate the effectiveness of our proposed architecture for handling complex PDE systems, highlighting its superior performance and generalization ability in cross-task PDE modeling.

In the fine-tuning stage, we conduct 200 and 500 epochs of fine-tuning on each dataset. The results show that after 500 epochs, our model achieved state-of-the-art performance on 9 out of $\mathcal{L}_2$ tasks, surpassing advanced pre-trained models on the majority of tasks. Compared with training from scratch, fine-tuning on pretrained weights generally leads to better performance; moreover, increasing the number of fine-tuning steps typically yields higher prediction accuracy. These results demonstrate the superior performance of our proposed model on sparse datasets, highlighting its stronger generalization ability and adaptability.

In summary, our model demonstrates significant advantages in operator learning for PDE tasks. With the aid of fine-tuning strategies, it can rapidly adapt to specific tasks and achieve 10 global best performances across 12 benchmark datasets, highlighting its strong modeling capability in capturing complex dynamics and multi-scale features, as well as its excellent transferability.

Table 1: The experiments are divided into two parts: one reports the pre-training performance of the model, and the other shows the fine-tuning results on each task. Here, "-200" denotes fine-tuning for 200 epochs, and "-500" for 500 epochs. The evaluation metric is the $\mathcal{L}_2$ loss. The best result within each part is highlighted in **bold**, while the overall best result is emphasized in blue bold.

| L2RE | FNO-$\nu$ | | | PDEBench CNS-$(\eta, \zeta)$, DR, SWE | | | | | | | | PDEArena | | CFDBench |
| Model | 1e-5 | 1e-4 | 1e-3 | 1,0.1 | 1,0.01 | M1 | 0.1,0.1 | 0.1,0.01 | M0.1 | DR | SWE | NS | NS-cond | - |
|---|---|---|---|---|---|---|---|---|---|---|---|---|---|---|
| **Pre-trained** | | | | | | | | | | | | | | |
| FNO | 0.116 | 0.0922 | 0.0156 | 0.151 | 0.108 | 0.130 | 0.230 | 0.076 | 0.153 | 0.0321 | 0.0091 | 0.210 | 0.384 | 0.0274 |
| UNet | 0.198 | 0.119 | 0.0245 | 0.334 | 0.291 | 0.313 | 0.569 | 0.357 | 0.463 | 0.0971 | 0.0521 | 0.102 | 0.337 | 0.209 |
| FFNO | 0.121 | 0.0503 | 0.0099 | 0.0212 | 0.052 | 0.0366 | 0.162 | 0.0452 | 0.104 | 0.0571 | 0.0116 | 0.0839 | 0.602 | 0.0071 |
| GK-T | 0.134 | 0.0792 | 0.0098 | 0.0341 | 0.0377 | 0.0359 | 0.0274 | 0.0366 | 0.0320 | 0.0359 | 0.0069 | **0.0952** | 0.423 | 0.0105 |
| GNOT | 0.157 | **0.0443** | 0.0125 | 0.0325 | 0.0420 | 0.0373 | 0.0228 | 0.0341 | 0.0285 | 0.0311 | 0.0068 | 0.172 | **0.325** | **0.0088** |
| Oformer | 0.1705 | 0.0645 | 0.0104 | 0.0417 | 0.0625 | 0.0521 | 0.0254 | 0.0205 | 0.0229 | 0.0192 | 0.0072 | 0.135 | 0.332 | 0.0102 |
| MPP | - | - | - | - | - | 0.0442 | - | - | 0.0312 | **0.0168** | 0.0066 | - | - | - |
| DPOT | **0.0976** | 0.0606 | 0.00954 | 0.0173 | 0.0397 | 0.0285 | 0.0132 | 0.0220 | 0.0176 | 0.0321 | 0.0056 | 0.125 | 0.384 | 0.0095 |
| Ours | 0.1195 | 0.0951 | **0.0093** | **0.0167** | **0.0373** | **0.0270** | **0.0120** | **0.0202** | **0.0161** | 0.0308 | **0.0052** | 0.132 | 0.409 | 0.0112 |
| **FineTune** | | | | | | | | | | | | | | |
| DPOT-FT200 | 0.0511 | 0.0431 | 0.0073 | 0.0136 | 0.0238 | 0.0187 | 0.0168 | 0.0145 | 0.0157 | 0.0194 | 0.0028 | 0.103 | 0.313 | 0.0054 |
| Ours-FT200 | 0.0581 | 0.0313 | 0.0056 | 0.0139 | 0.0182 | 0.0161 | 0.0155 | 0.0112 | 0.0134 | 0.0198 | 0.0032 | 0.0793 | 0.321 | 0.0045 |
| DPOT-FT500 | 0.0520 | 0.0367 | 0.0058 | 0.0112 | 0.0195 | 0.0153 | 0.0174 | 0.0138 | 0.0156 | 0.0148 | **0.0024** | 0.0910 | **0.280** | 0.0039 |
| Ours-FT500 | **0.0505** | **0.0217** | **0.0043** | **0.0094** | **0.0134** | **0.0114** | **0.0123** | **0.0083** | **0.0103** | **0.0117** | 0.0026 | **0.0683** | 0.285 | **0.0038** |

## 4.3 DOWNSTREAM TASKS EXPERIMENTS

To evaluate the generalization and transferability of our model, we conducted downstream experiments on a two-dimensional high-resolution turbulence task. In these experiments, we reused most of the parameters from the pre-trained model, including the weights of the MoE modules and the spatio-temporal encoding. The visualization of the model predictions is shown in Fig. 4.

As illustrated in Fig. 3, most models fine-tuned from pre-trained weights outperform those trained from scratch, which demonstrates the effectiveness of large-scale pre-training. This indicates that the model can acquire generalizable PDE knowledge and successfully transfer it to specific downstream tasks. On the two-

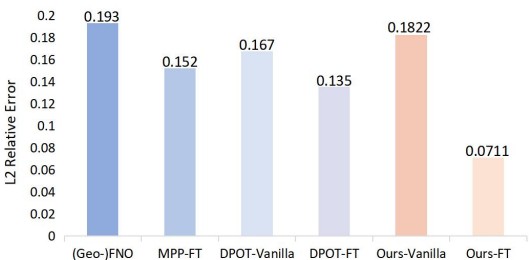

Figure 3: Performance comparison of different models on the 2D high-resolution turbulence task. The evaluation metric is the $\mathcal{L}_2$ relative error, where Vanilla denotes training from scratch, and -FT indicates results after 500 fine-tuning epochs on the downstream task.

dimensional high-resolution turbulence task, our model achieves a 47.3% improvement in prediction accuracy, reaching the best performance. These results show that our pre-trained model learns more effective representations and can be successfully transferred to downstream tasks with only limited fine-tuning, highlighting its advantage in capturing PDE-specific features.

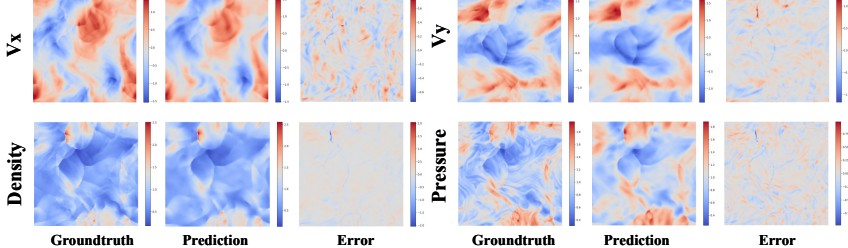

Figure 4: Visualization of 2D high-resolution turbulence prediction results. (1) The first column shows the true values, the second column shows the model predictions, and the third column shows the corresponding errors. (2) The predicted physical quantities are horizontal velocity, vertical velocity, density field, and pressure field.

Table 3: Ablation experiments of our proposed model on the PDEBench datasets. "w/o" denotes the removal of the corresponding component.

| Method | 1,0.1 | 1,0.01 | 0.1,0.1 | 0.1,0.01 | DR | SWE | Avg L2 | Promotion |
|---|---|---|---|---|---|---|---|---|
| Ours | 0.0144 | 0.0355 | 0.0135 | 0.0178 | 0.0282 | 0.0045 | 0.0173 | **-** |
| w/o Sub-MoE | 0.0157 | 0.0393 | 0.0130 | 0.0209 | 0.0245 | 0.0049 | 0.0197 | **0.0024** |
| w/o Load Balance Loss | 0.0135 | 0.0335 | 0.0109 | 0.0159 | 0.0265 | 0.0062 | 0.0178 | **0.0005** |
| FlashAttn + AFNO Sum | 0.0149 | 0.0363 | 0.0136 | 0.0178 | 0.0304 | 0.0046 | 0.0196 | **0.0023** |

### 4.4 SCALING EXPERIMENTS

The number of experts in the MoE is a key factor affecting the performance of pre-trained models. We fix the number of experts activated each time, vary the number of unshared experts, and use the average loss $\mathcal{L}_2$ between datasets as the evaluation metric to study the impact of the number of experts on pre-trained model performance. In selected datasets, we set three training strategies: Zero-shot, FT-200 (fine-tuning for 200 steps), and FT-500 (fine-tuning for 500 steps). The results, shown in Table 2, show that for specific tasks, fine-tuning the pre-trained model can significantly improve performance, and more rounds

Table 2: The impact of the number of experts on performance.

| Setting | Num | FNO | PDEBench | SWE | Avg L2 |
|---|---|---|---|---|---|
| Zero-shot | 2 | 0.0625 | 0.0332 | 0.0057 | 0.0338 |
| | 4 | 0.0615 | 0.1974 | 0.0035 | 0.0875 |
| | 6 | 0.0635 | 0.2600 | 0.0029 | 0.1088 |
| | 12 | 0.0630 | 0.2593 | 0.0030 | 0.1084 |
| FT-200 | 2 | 0.0575 | 0.0182 | 0.0024 | 0.0262 |
| | 4 | 0.0563 | 0.0150 | 0.0025 | 0.0246 |
| | 6 | 0.0577 | 0.0240 | 0.0579 | 0.0466 |
| | 12 | 0.0575 | 0.1896 | 0.0025 | 0.0832 |
| FT-500 | 2 | 0.0519 | 0.0126 | 0.0022 | 0.0222 |
| | 4 | 0.0504 | 0.0114 | 0.0025 | 0.0214 |
| | 6 | 0.0512 | 0.0165 | 0.0026 | 0.0234 |
| | 12 | 0.0520 | 0.0144 | 0.0025 | 0.0230 |

of fine-tuning lead to better results. For complex MoE architectures, more experts are not necessarily better; increasing the number of experts makes optimization more difficult and resource allocation more complex. For different tasks, there exists an optimal range for the number of experts, and choosing the right number is crucial to fully realizing the performance of the MoE model.

### 4.5 ABLATION STUDIES

To validate the effectiveness of our model, we conducted experiments on six sub-tasks of the PDEBench dataset to assess the impact of different modules on model performance. Using the complete model as the baseline, we systematically performed ablation studies by progressively removing or replacing key modules, with the average $\mathcal{L}_2$ error (Avg. $\mathcal{L}_2$) serving as the primary comprehensive evaluation metric. The results are shown in Table 3.

**Impact of Sub-MoE:** Removing the Sub-MoE module led to an increase of 0.0024 in the average $\mathcal{L}_2$ error. Among all modules, Sub-MoE contributed most significantly to performance improvement, indicating that it plays an important role in effectively capturing multi-scale and diverse features, thereby fully validating its importance.

**Impact of the load balancing loss:** Removing the load balancing loss resulted in an increase of 0.0005 in the average $\mathcal{L}_2$ error. Although its contribution is smaller compared to other modules, it still provides a certain improvement to model performance.

**Impact of the fusion strategy between AFNO and FlashAttention:** Changing the fusion of AFNO and FlashAttention from MoE to simple addition increases the Avg. $\mathcal{L}_2$ error by 0.0023. This demonstrates that our nested MoE can select the most suitable expert for different inputs, thereby enhancing model performance and generalization ability, and validates the rationality of the design.

## 5 CONCLUSION

This paper proposes a large-scale PDE pre-trained neural operator based on a nested Mixture-of-Experts (MoE) architecture. We design the nested MoE framework, which consists of image-level MoE and token-level MoE, and conduct extensive training on twelve PDE datasets to obtain a universal pre-trained model. Our model successfully transfers to specific tasks and new downstream tasks, achieving state-of-the-art performance on most datasets. Furthermore, this paper explores the suitability and advantages of MoE architectures for large-scale PDE pre-trained neural operators, introducing MoE into this field for the first time and revealing new potential for solving PDEs.

## 6 ETHICS STATEMENT

This research adhered to ICLR's ethical guidelines and did not involve any human subjects or animal experiments. All datasets used adhered to relevant privacy guidelines and were confidential. We took every effort to minimize potential bias and avoid discriminatory results. No personally identifiable information was used, and no experiments were performed that could raise privacy or safety concerns. We are committed to transparency and integrity throughout our research.

## 7 REPRODUCIBILITY STATEMENT

Details of the training procedure, model configuration, hardware environment, and datasets are provided in Appendices A.2 and A.3. All datasets used are publicly available, and the source code will be released upon acceptance of the paper.

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

# A APPENDIX

## A.1 LLM USAGE

During the manuscript writing and revision process, we used a Large Language Model (LLM) to assist. Specifically, LLM was used to improve the accuracy and readability of the language, and to help ensure the overall structure and clarity of the paper. This tool primarily assisted with tasks such as sentence reconstruction, grammatical proofreading, and improving text coherence.

## A.2 EXPERIMENTAL DETAILS

**Pre-training.** We pre-trained the model on 8 NVIDIA RTX 4090 GPUs using the Adam optimizer with an initial learning rate of $1.0 \times 10^{-3}$ and a cyclic learning rate schedule (cycle), including 200 warm-up epochs. The total training lasted 1000 epochs with a batch size of 32. To mitigate the effects of varying dataset sizes, training weights were assigned to each dataset. During training, we used $T = 10$ time steps to predict the next frame, maintaining consistency with the original settings of most datasets.

**Fine-tuning.** In the fine-tuning stage, we loaded the pre-trained weights and performed 200-epoch and 500-epoch fine-tuning on each subset. The key module of the model is the nested MoE layer, whose parameters are shared across different frequency components along the channel dimension, enabling cross-level expert collaboration.

## A.3 DETAILED INFORMATION OF DATASETS

We list the configurations of the PDE datasets used for pre-training along with detailed descriptions of the governing partial differential equations:

Table 4: Train and test set sizes of the PDE datasets used for pre-training.

| | FNO-$\nu$ | | | PDEBench CNS-($\eta, \zeta$), DR, SWE | | | | | | PDEArena | | CFDBench |
| | 1e-5 | 1e-4 | 1e-3 | 1,0.1 | 1,0.01 | 0.1,0.1 | 0.1,0.01 | DR | SWE | NS | NS-cond | - |
|---|---|---|---|---|---|---|---|---|---|---|---|---|
| Train set size | 100 | 9800 | 1000 | 9000 | 9000 | 9000 | 9000 | 900 | 900 | 5200 | 2496 | 9000 |
| Test set size | 200 | 200 | 200 | 1000 | 1000 | 1000 | 1000 | 100 | 100 | 1300 | 600 | 1000 |

- **FNO-$v$**: This dataset focuses on the temporal evolution of the two-dimensional incompressible fluid vorticity field $w(x, t)$, where $(x, t) \in [0, 1]^2 \times [0, T]$. The dynamics are governed by the two-dimensional Navier–Stokes equations in the vorticity–streamfunction formulation:

$$\partial_t w + u \cdot \nabla w = \nu \Delta w + f(x), \qquad \nabla \cdot u = 0, \tag{17}$$

where $u$ denotes the velocity field, $\nu$ is the viscosity coefficient, $\Delta$ represents the Laplace operator, and $f(x)$ denotes the external forcing term. By varying the viscosity $\nu$, the dataset provides fluid dynamics simulations under different flow regimes, enabling the study of how viscosity influences the evolution of vortex structures.

- **PDEBench-CMS**: This dataset focuses on the numerical simulation of compressible fluid mechanics (CMS). The goal is to predict the temporal evolution of the velocity field $u(x, t)$, the pressure field $p(x, t)$, and the density field $\rho(x, t)$ over the spatio-temporal domain $(x, t) \in [0, 1]^2 \times [0, 1]$. The data are generated based on the governing equations of compressible fluid dynamics, which consist of the conservation of mass, momentum, and energy: **Mass conservation (continuity equation):**

$$\partial_t \rho + \nabla \cdot (\rho u) = 0, \tag{18}$$

$$\rho \left( \partial_t u + u \cdot \nabla u \right) = -\nabla p + \eta \Delta u + \left( \zeta + \frac{\eta}{3} \right) \nabla (\nabla \cdot u), \tag{19}$$

$$\partial_t \left( \frac{3}{2} p + \frac{\rho u^2}{2} \right) = -\nabla \cdot \left[ \left( \varepsilon + p + \frac{\rho u^2}{2} \right) u - u \cdot \sigma' \right], \tag{20}$$

where $\eta$ denotes the shear viscosity coefficient and $\zeta$ the bulk viscosity coefficient. $\varepsilon$ is the energy density and $\sigma'$ is the stress tensor.

- **PDEBench-SWE**: The dataset is derived from PDEBench and focuses on the numerical simulation of the Shallow Water Equations (SWE). The objective is to predict the water depth field $h(x,t)$ over the spatiotemporal domain $(x,t) \in [-1,1]^2 \times [0,5]$. The SWE is a set of approximate governing equations widely used in ocean dynamics, flood modeling, and geomorphological evolution studies. The governing equations are given as follows:

$$\partial_t h + \nabla \cdot (hu) = 0, \tag{21}$$

$$\partial_t (hu) + \nabla \cdot \left(\tfrac{1}{2}hu^2 + \tfrac{1}{2}grh^2\right) = -grh\nabla b, \tag{22}$$

- **PDEBench-DR**: The dataset is derived from PDEBench and focuses on the numerical simulation of diffusion–reaction (DR) systems. The objective is to predict the density field $u(x,t)$ over the spatiotemporal domain $(x,t) \in [-2.5, 2.5]^2 \times [0,1]$. The governing equation is given by:

$$\partial_t u = D\nabla^2 u + R(u), \tag{23}$$

where $D$ is the diffusion coefficient and $R(u)$ denotes the nonlinear reaction term.

- **PDEArena-NS1/2**: The dataset is derived from PDEArena and focuses on the numerical simulation of incompressible Navier–Stokes (NS) flows. The objective is to predict the velocity field $u(x,t)$, pressure field $p(x,t)$, and density field $\rho(x,t)$ over the spatiotemporal domain $(x,t) \in [0,32]^2 \times [0,24]$. The governing equations are given as follows:

$$\partial_t v = -v \cdot \nabla v + \mu \nabla^2 v - \nabla p + f, \tag{24}$$

$$\nabla \cdot v = 0, \tag{25}$$

where $v$ denotes the velocity field, $\mu$ is the viscosity coefficient, $p$ is the pressure, and $f$ represents external forcing.

- **CFDBench**: The dataset is derived from CFDBench and focuses on the numerical simulation of incompressible or weakly compressible flows in irregular geometries. The objective is to predict the velocity field $u(x,t)$ and the pressure field $p(x,t)$ over domains with complex boundaries. The governing equations are given as follows:

$$\partial_t (\rho u) + \nabla \cdot (\rho u^2) = -\nabla p + \nabla \cdot \mu(\nabla u + \nabla u^\top), \tag{26}$$

$$\nabla \cdot (\rho u) = 0, \tag{27}$$

where $\rho$ is the fluid density, $u$ is the velocity field, $p$ is the pressure, and $\mu$ denotes the viscosity coefficient.

### A.4 VISUALIZATION

For each specific subtask, we first load the model weights pretrained on large-scale PDE datasets, and then fine-tune the model for each subtask to fully leverage the general features and structural information learned during pretraining. During fine-tuning, the model can adapt to the data distribution and equation characteristics of different subtasks, thereby improving prediction accuracy and generalization capability. The visualization of the prediction results is shown in the figure. For each data series, we select one representative equation to illustrate the model's performance across different tasks. These visualizations allow us to observe the model's ability to capture spatiotemporal trends, local details, and global patterns, and facilitate comparison with other baseline methods, thereby demonstrating the effectiveness and advantages of pretrained weights in downstream tasks.

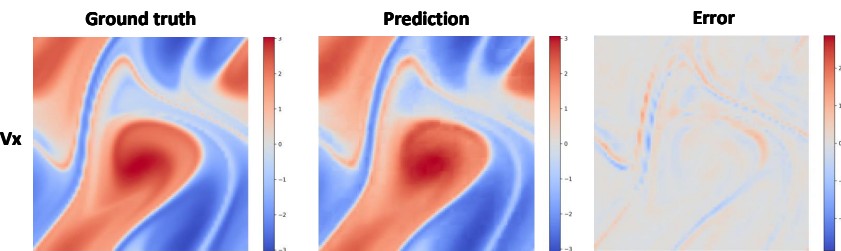

Figure 5: FNO series of result visualizations. (1) The first column shows the true value, the second column shows the model prediction value, and the third column shows the corresponding error. (2) Each row is the predicted physical quantity.

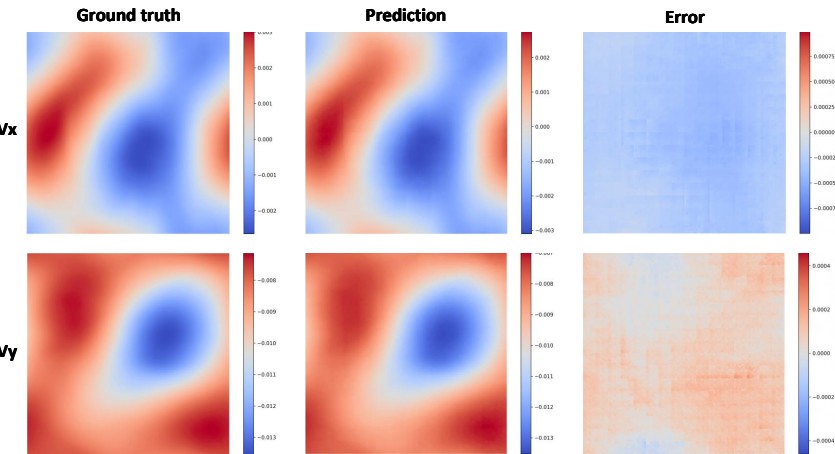

Figure 6: PDEBench series of result visualizations. (1) The first column shows the true value, the second column shows the model prediction value, and the third column shows the corresponding error. (2) Each row is the predicted physical quantity.

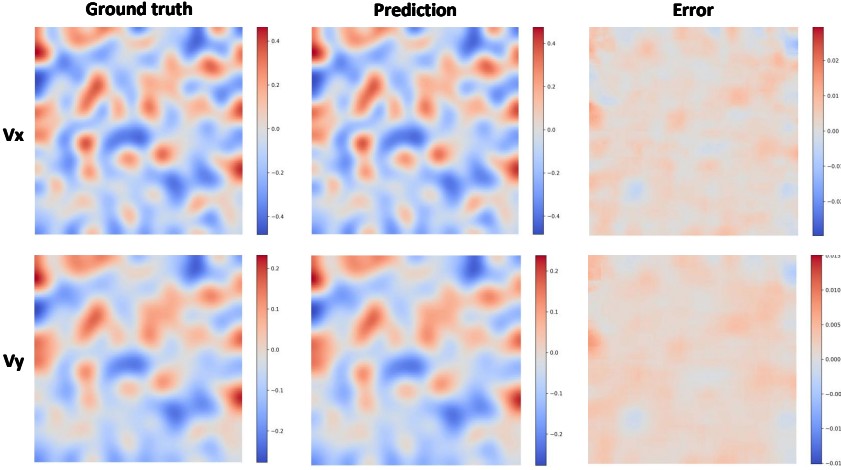

Figure 7: DR series of result visualizations. (1) The first column shows the true value, the second column shows the model prediction value, and the third column shows the corresponding error. (2) Each row is the predicted physical quantity.

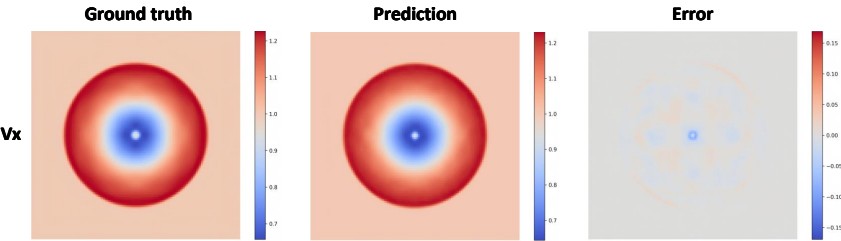

Figure 8: SWE series of result visualizations. (1) The first column shows the true value, the second column shows the model prediction value, and the third column shows the corresponding error. (2) Each row is the predicted physical quantity.

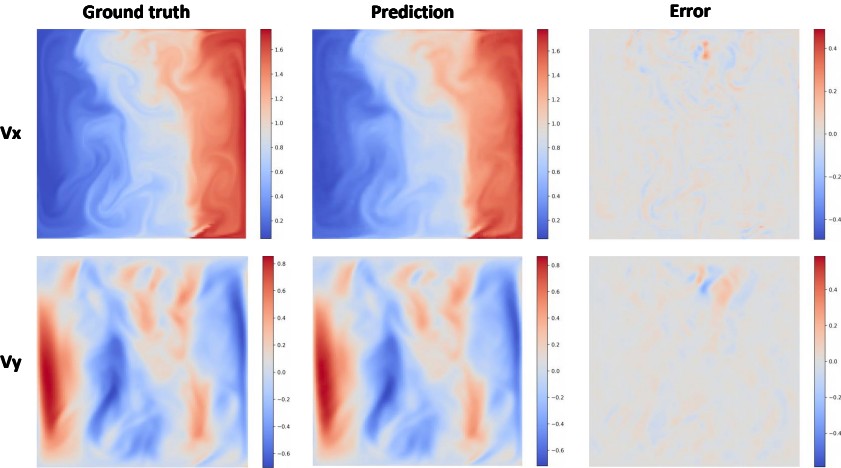

Figure 9: PDEArena series of result visualizations. (1) The first column shows the true value, the second column shows the model prediction value, and the third column shows the corresponding error. (2) Each row is the predicted physical quantity.

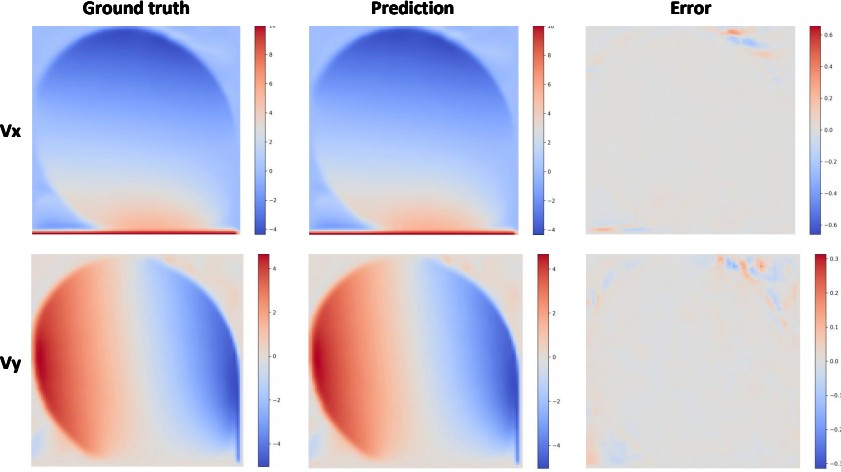

Figure 10: CFDBench series of result visualizations. (1) The first column shows the true value, the second column shows the model prediction value, and the third column shows the corresponding error. (2) Each row is the predicted physical quantity.

