# OpenReview forum: "Nestor: A Nested MOE-based Neural Operator for Large-Scale PDE Pre-Training"
_ICLR.cc/2026/Conference — ICLR 2026 Conference Withdrawn Submission_

### Official Review · Reviewer_Vrjs · 2025-10-23

**Soundness:** 2
**Presentation:** 2
**Contribution:** 2
**Rating:** 2
**Confidence:** 4

**Summary:**

This paper presents NESTOR, a large-scale pre-trained neural operator based on a nested Mixture-of-Experts (MoE) framework. It designs a image-level MoE to capture global dependencies and a token-level Sub-MoE for local dependencies. A large-scale pre-training experiment is conducted on 12 PDE datasets collected by DPOT to evaluate the model.

**Strengths:**

- The paper is overall easy to follow. The idea is easy to understand.
- The model architecture design is overall well motivated. The MoE design is a reasonable approach for large-scale heterogeneous PDE datasets pre-training.
- The experimental results demonstrates the architecture's effectiveness over existing methods. The fine-tuning experiments are also interesting and show the model's transfer learning capability.

**Weaknesses:**

- It seems that this paper is finished in a hurry without careful proofreading. In Figure 1 "PDF's Diversity" "PDF's Complexity", which I think it should be "PDE". The citation formatting is also irregular throughout this paper.

- Though designed carefully, the technical contributions regarding the model architecture and the task loss are somewhat limited.

- It seems that the performance gain over the baseline is relatively modest; 6 of 14 metrics of your model does not exceed the baseline. As the MoE design targets better processing of diverse and complex PDE datasets, this results is not promising.

- Nor have you reported the model parameter numbers, as DPOT has provided results of different scales, so comparison of models with different scales of parameters is not fair. A recently published paper "Unisolver" [1] also includes DPOT as its baseline and achieves better results. You should consider compare with it as well.

  [1] Hang, Zhou, et al. "Unisolver: PDE-Conditional Transformers Towards Universal Neural PDE Solvers." *ICML 2025*. 2025.

**Questions:**

See weaknesses.

---

### Official Review · Reviewer_cnh5 · 2025-10-29

**Soundness:** 2
**Presentation:** 2
**Contribution:** 2
**Rating:** 2
**Confidence:** 4

**Summary:**

This paper introduces NESTOR, a nested Mixture-of-Experts (MoE) neural operator designed for large-scale PDE pretraining. The model combines an image-level MoE (for global PDE diversity) with a token-level Sub-MoE (for local spatial complexity), integrating AFNO and FlashAttention experts under hierarchical routing. Experiments on twelve PDE datasets show strong in-distribution performance.

**Strengths:**

- This work presents one of the first successfully trained Mixture-of-Experts (MoE) architectures in the context of foundation models for PDEs, marking a valuable step toward scalable, modular operator learning.

- Despite the complexity of the nested MoE design (image-level and token-level routing), the authors demonstrate that the model can be trained stably and efficiently across a large and diverse collection of PDE datasets.

- The paper shows attention to optimization stability, incorporating dedicated load-balancing losses at both MoE levels to ensure fair expert utilization and prevent collapse.

- The authors include targeted ablation studies examining the contribution of key components, namely the Sub-MoE, the load-balancing term, and the fusion strategy between AFNO and FlashAttention, providing at least partial insight into which parts of the architecture matter most.

**Weaknesses:**

- The paper does not reference or compare against current state-of-the-art foundation models for PDEs, such as Poseidon [1], which already explore large-scale pretraining and transfer to unseen downstream tasks.

- In PDE settings, the concept of a temporal “frame” (Eq. 2) is ambiguous. A specific time step Δt must be fixed to define the “next frame,” which makes the model resolution-dependent. Once the temporal resolution is changed or sub-sampled, the model becomes inapplicable. Moreover, different PDEs evolve on very different temporal scales, so mixing them within one model is problematic and not clearly addressed.

- The authors claim that Transformers “struggle to effectively represent kernel integral operators” (lines 204–205), which is inaccurate. Several recent works [1, 2, 3] demonstrate that Transformers can approximate such operators effectively.

- The core motivation of a foundation model is largely missed in this paper. Although the authors perform large-scale pretraining on a mixture of PDE datasets, the model is then evaluated and fine-tuned on exactly the same distributions. This setup does not align with the goal of foundation models, which is to learn broadly transferable representations that can generalize to new, unseen tasks and data domains. In real-world scientific and engineering problems, new data will almost never follow the same distribution as the pretraining datasets, so true foundation models must demonstrate out-of-distribution transfer and robustness to domain shifts. Here, the model is fine-tuned only on datasets that were already part of the pretraining pool, which effectively measures continued training rather than transfer learning. By contrast, prior works such as Poseidon [1] explicitly evaluated pretraining on one set of PDEs and transfer to multiple unseen downstream tasks, thereby revealing both the strengths and limitations of generalization. In the current paper, only a single transfer-learning experiment (the turbulence task) is provided, which is insufficient to demonstrate general-purpose or cross-physics generalization. Much stronger evidence, such as systematic transfer to unseen PDE types, new geometries, or distinct physical regimes, would be needed to substantiate the claim that the proposed MoE architecture contributes meaningfully toward foundation models for PDEs.

- There is no study of scaling behavior with respect to model size or data size at the transfer-learning level. Such experiments are essential for assessing the efficiency and scalability of foundation models. Only one scaling experiment with respect to data size (on the pretraining dataset level) is provided.

- Ablation studies on expert-utilization statistics are missing. It remains unclear whether the nested MoE routing actually utilizes multiple experts or collapses to a single expert. Quantitative utilization metrics would be necessary to understand the model’s internal dynamics and validate the effectiveness of the MoE design.

- Figure 1 is very cluttered and hard to follow. It is unclear where the token-level and image-level experts are applied and how they are fused. The distinction between “shared,” “non-shared,” and “Sub-MoE” experts is also unclear. A clearer architectural figure with labeled modules (perhaps merging Figures 1 and 2) would significantly improve readability. Additionally, in the section 3.3, the notation for H and W is confusing: sometimes these represent original image dimensions, and other times latent dimensions after patchification.

- The authors never report:

(1) total parameter count,

(2) active parameters per token/image

(3) memory/computation cost relative to dense baselines (e.g., FNO, DPOT).

Even though the overall direction of this work is promising and the proposed nested MoE design could be an interesting step toward scalable PDE modeling, the current experimental evidence is far from sufficient to support its usefulness in true foundation model settings

___

[1] Herde, M., Raonic, B., Rohner, T., Käppeli, R., Molinaro, R., de Bézenac, E., & Mishra, S. (2024). Poseidon: Efficient foundation models for pdes. Advances in Neural Information Processing Systems, 37, 72525-72624.

[2] Li, Z., Meidani, K., & Farimani, A. B. (2022). Transformer for partial differential equations' operator learning. arXiv preprint arXiv:2205.13671.

[3] Ovadia, O., Kahana, A., Stinis, P., Turkel, E., Givoli, D., & Karniadakis, G. E. (2024). Vito: Vision transformer-operator. Computer Methods in Applied Mechanics and Engineering, 428, 117109.

**Questions:**

- How does the nested routing mechanism behave when multiple PDE types exhibit overlapping spatial or temporal statistics—does the router consistently assign similar experts, or does it learn more nuanced distinctions?

- To what extent do the image-level and token-level experts specialize in different physical regimes or frequency ranges, and is there any interpretable pattern behind this specialization?

- Since the model is pretrained and fine-tuned on the same datasets and distributions, how can we interpret its claimed generalization ability within the broader context of foundation models?

- Given that no statistics on expert utilization are reported, how can we be confident that the nested MoE architecture does not collapse to a single active expert during training?

- Would the nested MoE design still maintain stable expert utilization if trained on a single-PDE dataset, or does its benefit mainly arise in multi-PDE pretraining scenarios?

**Details Of Ethics Concerns:**

/

---

### Official Review · Reviewer_s8iH · 2025-10-31

**Soundness:** 3
**Presentation:** 2
**Contribution:** 2
**Rating:** 2
**Confidence:** 4

**Summary:**

This paper introduces NESTOR, a framework based on a Mixture of Experts for solving Partial Differential Equations. Through hierarchical routing at the image and token levels, the model aims at capturing variation in PDE solution data at different scales. Moreover, the framework benefits from large-scale PDE pre-training, followed by fine-tuning on specific PDEs and sets of parameters;  enjoying very high performance on next-step prediction for transient simulations. Finally, ablation studies showcase the importance of the various architectural blocks introduced in this model.

**Strengths:**

-	Introduction of an innovative Mixture of Experts architecture to deal with image level and token level PDE solution complexity
-	Use of a pre-training procedure on a vast array of PDEs and parameters, with fine-tuning on particular instances leading to increased performance.
-	State of the art results obtained on multiple PDE benchmarks studied
-	Ablation studies justify the need for the various components of the framework.

**Weaknesses:**

-	The model is evaluated only on next-step prediction accuracy, without consideration for rollout stability over longer horizons. In contrast, DPOT incorporated noise during training to enhance robustness. Without similar analysis, it remains unclear whether this model maintains stability during multi-step predictions.
-	The model lacks interpretability regarding expert behavior. It is not demonstrated whether the individual experts specialize in different physical regimes or scales, leaving uncertain whether the hierarchical routing effectively captures multi-scale interactions.
-	The proposed architecture may face scalability challenges for high-resolution meshes, where the number of spatial tokens N=H×W exceeds 10^6. The paper does not discuss strategies for handling the associated computational and memory bottlenecks.
-	Increasing the number of experts appears to yield only limited improvements in performance. This raises questions about the efficiency of expert utilization and whether the architecture fully exploits the Mixture-of-Experts capacity.
-	The PDE formulations in the appendix omit key details such as initial conditions, and the figures do not specify the time steps at which the visualizations are generated. These omissions hinder reproducibility and interpretation of the results.
-	Some claims are vague, particularly those contrasting this work with DPOT. The authors should clearly articulate what specific limitations of DPOT are addressed here, as the described pretraining procedure appears conceptually similar.
-	 Typo l.369:  “9 out of L2 tasks” -> 9 out of 12 tasks ?

**Questions:**

-	Have the authors considered the stability of the model predictions in the context of longer auto-regressive rollouts ?
-	In table 2, when the authors state 2 experts are used, are those experts the shared expert and one unshared expert ?
-	Could the authors precise which initial conditions are used and times are represented in the PDEs and solutions plotted in the appendix ?
-	How does the current pre-training overcome the limitations of the DPOT pre-training.

---

### Official Review · Reviewer_ScpY · 2025-11-11

**Soundness:** 4
**Presentation:** 4
**Contribution:** 4
**Rating:** 8
**Confidence:** 5

**Summary:**

The paper proposes  a nested mixture-of-experts (MoE) neural operator for large-scale PDE pre-training. It nests an image-level MoE with a token-level Sub-MoE (to model local, fine-grained dependencies inside each field). A spatio-temporal patch encoder feeds experts; AFNO serves as a shared global-frequency expert; FlashAttention is used within the transformer blocks; and two load-balancing losses regularize routing at both MoE levels. The model is pre-trained on 12 PDE datasets (mixture of FNO, PDEBench, PDEArena and CFDBench sources) with a next-frame objective, then fine-tuned. Results show SOTA or strong performance on 6/12 tasks in pre-training, and after fine-tuning SOTA on 9 tasks.

**Strengths:**

1. The idea is novel. It uses a two-level MoE for PDE operators, which is a novel architectural twist for scientific operators. Besides, using AFNO as a shared global expert plus FlashAttention inside an MoE operator is a thoughtful combination.

2. The paper is clear and well-organized, and the figures are quite good and easy to read.

3. The overall performance is positive to show the effectiveness of the method.

**Weaknesses:**

1. Computational cost not quantified. MoE + FlashAttention + AFNO is likely compute/memory heavy. The paper lacks FLOPs/throughput, GPU hours, tokens/updates, or cost-vs-quality curves, and no comparison to strong single-backbone operators at equal compute.

2. Most experiments appear on regular-grid, 2D fields with autoregressive next-frame targets. Little is shown for 3D, irregular meshes, varying boundary/geometry conditions.

**Questions:**

None

---

### Note · Authors · 2025-11-12

I have read and agree with the venue's withdrawal policy on behalf of myself and my co-authors.